# A 3D CBCT Analysis of Airway and Cephalometric Values in Patients Diagnosed with Juvenile Idiopathic Arthritis Compared to a Control Group

Matthew Gibson [1], Randy Q. Cron [2], Matthew L. Stoll [2], Brian E. Kinard [3], Tessa Patterson [1] and Chung How Kau [1,*]

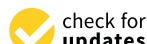



[1] Department of Orthodontics, University of Alabama at Birmingham, Birmingham, AL 35294, USA; mg308@uab.edu (M.G.); tep1101@uab.edu (T.P.)
[2] Department of Pediatric Rheumatology, University of Alabama at Birmingham, Birmingham, AL 35233, USA; randycron@uabmc.edu (R.Q.C.); mstoll@uabmc.edu (M.L.S.)
[3] Department of Oral and Maxillofacial Surgery, University of Alabama at Birmingham, Birmingham, AL 35294, USA; briankinard@uabmc.edu
* Correspondence: ckau@uab.edu

**Abstract:** Introduction: The temporomandibular joint (TMJ) is affected in 30–45% of juvenile idiopathic arthritis (JIA) patients, with all JIA subtypes at risk for TMJ involvement. JIA patients with TMJ involvement may present with altered craniofacial morphology, including micrognathia, mandibular retrognathia, a hyperdivergent mandibular plane angle, and skeletal anterior open bite. These features are also commonly present and associated with non-JIA pediatric patients with obstructive sleep apnea (OSA). Materials and Methods: The study was comprised of a group of 32 JIA patients and a group of 32 healthy control subjects. CBCT images were taken for all patients and were imported into Dolphin Imaging software. The Dolphin Imaging was used to measure the upper airway volumes and the most constricted cross-sectional areas of each patient. Cephalometric images were rendered from the CBCT data for each patient, and the following cephalometric values were identified: SNA angle, SNB angle, ANB angle, anterior facial height (AFH), posterior facial height (PFH), mandibular plane angle (SN-MP), FMA (FH-MP), overjet (OJ), and overbite (OB). Airway volumes, the most constricted cross-sectional area values, and cephalometric values were compared between the JIA and control groups. Results: For airway values, statistically significant differences were seen in the nasopharynx airway volume ($p = 0.004$), total upper airway volume ($p = 0.013$), and the most constricted cross-sectional area ($p = 0.026$). The oropharynx airway volume was not statistically significant ($p = 0.051$). For cephalometric values, only the posterior facial height showed a statistically significant difference ($p = 0.024$). Conclusions: There was a significant difference in airway dimensions in the JIA patients as compared to the control patients. In addition, the posterior facial dimensions seem to be affected in JIA patients. The ODDs ratio analysis further corroborated the findings that were significant.

**Keywords:** 3D airway analysis; 3D airway volumes; CBCT; orthodontics; oral and maxillofacial surgery

## 1. Introduction

Juvenile idiopathic arthritis (JIA) is characterized by the onset of arthritis of an unknown etiology before the age of 16 that persists for longer than a six-week duration. JIA is the most common rheumatic disease of childhood [1–3]. The progression and course of disease in JIA is unpredictable. It is self-limiting in some patients while in other circumstances it can be persistent with joint resorption. Depending on the timing of onset and severity of disease, JIA may cause growth disturbances, permanent joint damage, functional limitations, and short- or long-term disability [4].

The temporomandibular joint (TMJ) is one of the more frequently involved synovial joints in JIA patients and in some cases may be the only joint involved [5,6]. The TMJ is affected in 30–45% of JIA patients, with all JIA subtypes at risk for TMJ involvement [7,8]. Despite the high percentage of JIA patients that have TMJ arthritis and a high prevalence of associated morbidity, TMJ arthritis is an underdiagnosed component of JIA [9]. This is due in part to the challenges associated with traditional radiography and clinical examination of the TMJ, as well as the fact that TMJ symptoms are frequently absent or unreliable [9–12]. Even with radiographic evidence of condylar lesions, a significant percentage of JIA patients do not report any TMJ symptoms and may have a benign clinical TMJ examination [5,13,14]. Despite these challenges, early diagnosis and treatment of TMJ arthritis is critical, as the risk of craniofacial growth alterations is thought to increase in cases of earlier age of onset and results in longer duration of disease activity [9,15].

As TMJ involvement may be present without any clinical signs or symptoms, MRI is the gold standard for diagnosis of TMJ involvement [14]. This imaging technique is capable of detecting 63–91% of inflammatory changes, ranging from active arthritic changes as well as the sequelae of chronic arthritis [10]. Higher percentages of TMJ involvement are found when Gadolinium-enhanced magnetic resonance imaging (Gd-MRI) is employed, compared to ultrasound, computed tomography (CT), cone-beam-computed tomography (CBCT), or traditional radiography (i.e., panoramic radiographs) [5,11,16–18].

The anatomy of the TMJ makes it particularly susceptible to damage from arthritis [5]. Unlike other synovial joints, the primary center for mandibular growth is the mandibular condylar cartilage that lies under a thin layer of fibrocartilage on the head of the condyle [5]. Due to its superficial position, it is susceptible to damage or resorption in the presence of chronic inflammation of the TMJ, and subsequently, TMJ arthritis may have destructive effects on the growth of the mandible [19,20]. JIA patients with TMJ involvement may present with altered craniofacial morphology, including micrognathia, mandibular retrognathism, decreased total mandibular length, decreased posterior face heights, increased anterior face heights, skeletal anterior open bite, posterior mandibular rotation with hyperdivergent mandibular plane angle and occlusal plane angle, obtuse gonial angle, gonial notching, convex facial profile, and facial asymmetry with chin deviation in cases of unilateral joint destruction [5,6,13,15,21–27]. In one study, retrognathia was seen in 82% of JIA patients that had TMJ involvement diagnosed using the panoramic radiograph but was also seen in 55% of patients that did not show any signs of radiographic TMJ involvement. Those patients who did have TMJ involvement had a cephalometric A point-nasion-B point (ANB) angle that averaged 1.8 degrees greater than those without TMJ involvement [6]. It has been shown that even a small degree of condylar damage can be associated with significant alterations in craniofacial morphology [13].

In recent years, CBCT has become a commonly used imaging modality in dentistry and orthodontics. Compared to traditional CT, CBCT has a shorter acquisition time and a more focused radiation beam with less scatter, resulting in lower radiation dosages and making it more appropriate for routine clinical use in a dental setting [28]. CBCT data can be used to assess skeletal craniofacial features and hard tissue structural abnormalities of the TMJ and can also provide a method of volumetric, cross-sectional area, and linear-dimensional airway analysis.

Several 3D airway studies have shown smaller airway volumes and the most constricted cross-sectional area (MCCA) measurements in adults with OSA, as measured by CBCT, but less research has been conducted on pediatric populations [29–33]. Recently, a study was undertaken to compare the apnea hypopnea index (AHI) measured by polysomnography (PSG) to CBCT measures in pediatric OSA patients, and it was concluded that CBCT analysis may be a useful tool in the evaluation of the upper airway in pediatric OSA patients [34]. It was shown in both the nasopharynx and oropharynx that airway volume as well as mean and minimal cross-sectional area were smaller in patients with moderate to severe OSA (AHI > 5) compared to primary snorers (AHI < 1) that were matched for age, gender, and obesity [34]. Another study relating AHI to CBCT airway data

showed a statistically significant correlation between AHI and nasopharyngeal volume in children aged 7–11, as well as between AHI and the MCCA in the 12–17 age group [35].

Sleep-disordered breathing (SDB) and obstructive sleep apnea (OSA) may be among the functional impairments that JIA patients experience, which have been shown to be associated with pain, fatigue, and reduced health-related quality of life [36–38]. Daytime effects of OSA in children may include attention deficit, aggressive/impulsive behavior, hyperactivity, mood problems (including possibly depression/anxiety), poor school performance, headaches, fatigue, and excessive daytime sleepiness [39,40]. Although OSA is not caused by anatomic factors alone, certain craniofacial characteristics have been shown to have significant associations with OSA/SDB [41]. A narrow or retrusive maxilla, retrognathic mandible, hyperdivergent mandibular plane angle, steep palatal plane angle, increased lower face height, decreased ratio of posterior to anterior face heights, anterior open bite, obtuse gonial angle, obtuse cranial base angle, and inferior position of the hyoid bone have been shown to be associated with pediatric OSA [41–47]. Arthritic involvement of the TMJ and the resulting retrognathia/micrognathia might be related to the increased OSA prevalence in JIA patients, as both are common in JIA, and both are risk factors for OSA [48,49].

Attended PSG is one of the more predictable methods to diagnose OSA [39]. The prevalence of OSA determined by varying AHI thresholds in PSG in the general pediatric population is 1–4% [50]. Limited data exists to determine the prevalence of SDB/OSA problems in JIA patients, but it has been demonstrated that 40% of children with JIA had an AHI of greater than or equal to 1.5 events per hour, as measured by PSG [36,51]. Pain, fatigue, and medication side effects are often attributed to causing sleep disturbances in JIA patients without screening for underlying OSA, and the average time from the diagnosis of JIA to the discovery of OSA (when present) by PSG is 2.5 years, which may be significant given the complications of OSA in overall health and disease management [49]. Most JIA patients with OSA are undiagnosed and untreated, which likely contributes to a poorer treatment response and worse patient-reported outcomes [36].

As the potential craniofacial morphology alterations associated with TMJ arthritis in JIA patients closely resemble those commonly associated with airway compromise in pediatric patients, it is of particular interest to investigate potential airway-related sequalae of JIA. At present, there are no known published studies that evaluate upper airway dimensions in JIA patients using three-dimensional CBCT data. The purpose of this study is to evaluate upper airway volumes, the most constricted cross-sectional areas, and cephalometric values in JIA patients and compare them to healthy controls to determine if any differences exist.

## 2. Subject and Methods

This study is a retrospective case control study evaluating patients with and without JIA. Ethical approval was obtained from the University of Alabama Institutional Review Board #30000975. Patients presenting in the Orthodontic department at the University of Alabama School of Dentistry from 2018–2021 were reviewed. JIA patient had the following records: (1) MRI of the TMJ, (2) clinical photographs, and (3) a CBCT with a field of view that included the TMJ. A brief status of the JIA patients was also recorded. Patients who presented for orthodontic treatment and had clinical photographs and a CBCT were eligible as control patients. The control group was carefully selected to be age matched to the JIA group. Control patients were excluded from the study if they had any of the following characteristics: (1) history of arthritis, immune disease, or systemic disease confirmed by or under investigation by a rheumatologist, (2) history of diagnosed TMJ dysfunction, (3) congenital syndromes, or (4) craniofacial trauma.

The JIA study group comprised of 32 patients with a mean age of 13.6 years, consisting of 10 males and 22 females. The control group comprised 32 patients with a mean age of 13.75 years, consisting of 13 males and 19 females.

### 2.1. CBCT Acquisition

The CBCT image volumes for all JIA and control patients were captured using the same Carestream 9300 CBCT machine (Carestream Dental, Atlanta, GA, USA). The CBCT scan time was 14 s with a radiation dose of 20 μSi and image resolution up to 0.90 μm. Each image was saved in the universal Digital Imaging and Communications in Medicine or DICOM (*.dcm) format. Each image was imported into Dolphin Imaging software (Version 11.95 Premium, Dolphin Imaging & Management Solutions, Chatsworth, GA, USA). All images were oriented to the Frankfurt horizontal (FH) plane in the sagittal dimension and by leveling the right and left orbitale points in the frontal dimension.

### 2.2. Cephalometric Rendering and Analysis

Dolphin Imaging software was used to render a cephalometric image for each patient from the CBCT DICOM data. Each cephalometric image was generated from the algorithm proprietary to the imaging software and produced an orthogonal perspective with 0% magnification. The cephalometric image was digitally traced, and the following values were identified: sella-nasion-A point (SNA) angle, sella-nasion-B point (SNB) angle, A point-nasion-B point (ANB) angle, anterior facial height (AFH), posterior facial height (PFH), PFH to AFH ratio, mandibular plane angle (SN-MP), FMA (FH-MP), overjet (OJ), and overbite (OB). See Figure 1.

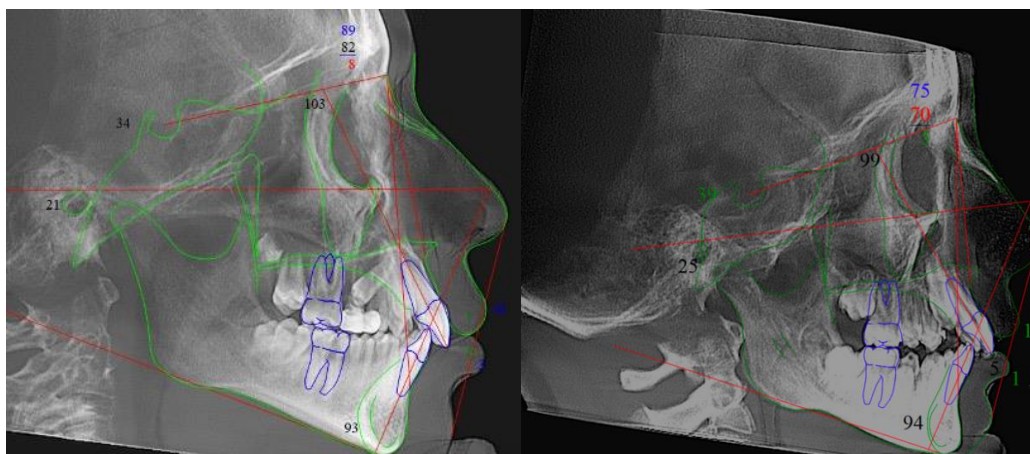

**Figure 1.** Cephalometric images rendered from CBCT DICOM data and traced using Dolphin Imaging. The (**left**) is a patient from the control group and the (**right**) is a patient from the JIA group.

### 2.3. Airway Measurements

The sinus/airway module of Dolphin Imaging was used to segment the three-dimensional representation of the upper airway, as well as determine the most constricted cross-sectional area (MCCA) of the upper airway. The inferior boundary of the oropharynx was set as a horizonal line from the anterior–superior corner of the third cervical vertebrae (C3), and the superior boundary was determined as a horizontal line from the posterior nasal spine (PNS). The nasopharynx extended from the superior limit of the oropharynx (PNS-horizontal) to a line extending vertically from PNS (PNS-vertical). The airway sensitivity in the Dolphin Imaging software was determined based on the most accurate visual rendering of the proper airway volume in our study groups and was set to 30 (on a scale from 0 to 100) for all patients. The MCCA was found within either the oropharynx or nasopharynx using the "minimal axial area" feature built into the Dolphin Imaging software. All measurements were done by a single investigator. A second investigator (MG) analyzed the airways of a random sample of 10 study patients. See Figure 2.

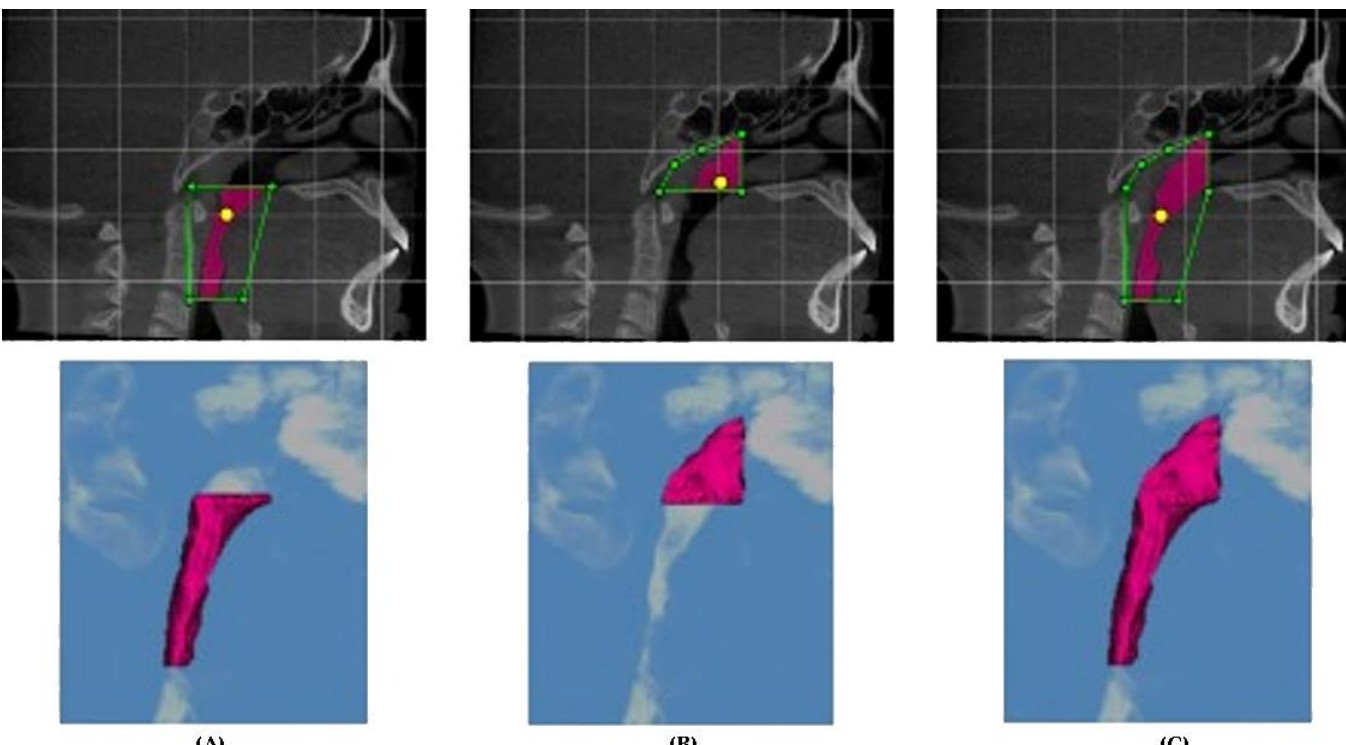

**Figure 2.** Showing boundaries of airway segmentation performed in Dolphin Imaging for (**A**) oropharynx, (**B**) nasopharynx, and (**C**) total upper airway.

### 2.4. Statistical Analysis

All data were entered onto an Excel spreadsheet. The datasets for each individual reading were tested and analyzed and found to be normally distributed. A Student's unpaired *t*-test was performed to compare airway volumes, MCCA values, and cephalometric values between the JIA patients and control group. In addition, the ODDs ratio was calculated. This analysis was used to determine the measure of association between a particular variable and the group measured.

### 3. Results

The results of each variable measured are presented on Table 1. Data from the first and second investigators were analyzed using a *t*-test, and no statistical differences were found. In addition, the data collected from each investigator had a high degree of reproducibility, indicating that the airway variables measured were reliable and accurate for analysis.

**Table 1.** Mean and standard deviation of each variable in control and JIA study groups, as well as *p*-value for statistical significance of unpaired Student *t*-test. In addition, the ODDS ratio was also calculated. * Indicates statistical significance.

| Variable | Control Mean (SD) | JIA Mean (SD) | *p*-Value Mean (SD) | ODDS Ratio (Confidence Range) |
|---|---|---|---|---|
| Age (years) | 13.75 (2.91) | 13.59 (2.73) | 0.825 | 1 |
| SNA (Degrees) | 83.06 (3.45) | 81.53 (4.27) | 0.12 | 1.00 (0.42–2.35) |
| SNB (Degrees) | 79.29 (3.65) | 78.43 (4.42) | 0.398 | 0.88 (0.32–2.44) |
| ANB (Degrees) | 3.69 (2.84) | 3.21 (2.15) | 0.448 | 0.45 (0.16–1.25) |
| Anterior Face Height(mm) | 113.12 (9.46) | 109.16 (8.51) | 0.083 | 0.72 (0.23–2.23) |
| Posterior Face Height (mm) | 75.39 (7.05) | 71.06 (7.87) | 0.024 * | 5.40 (1.66–17.56) * |
| PFH/AFH Ratio | 66.58 (5.67) | 65.12 (5.08) | 0.281 | 1.29 (0.48–3.44) |
| SN-MP (Degrees) | 32.45 (7.30) | 33.51 (6.12) | 0.529 | 0.85 (0.28–2.59) |

**Table 1.** *Cont.*

| Variable | Control Mean (SD) | JIA Mean (SD) | *p*-Value Mean (SD) | ODDS Ratio (Confidence Range) |
|---|---|---|---|---|
| FMA (Degrees) | 22.97 (7.33) | 22.88 (5.79) | 0.958 | 1 (0.34–2.97) |
| Overbite (mm) | 2.81 (2.37) | 2.13 (1.28) | 0.159 | 0.77 (0.15–3.53) |
| Overjet (mm) | 4.15 (2.33) | 4.43 (3.90) | 0.734 | 0.68 (0.25–1.84) |
| Oropharynx Airway Volume ($mm^3$) | 14,047.31 (5586.36) | 11,098.03 (6268.15) | 0.051 | 3.40 (1.18–9.81) * |
| Nasopharynx Airway Volume ($mm^3$) | 5827.03 (2425.85) | 4392.19 (1203.04) | 0.004 * | 4.59 (1.54–13.67) * |
| Total Upper Airway Volume ($mm^3$) | 19,874.34 (6790.35) | 15,490.22 (6945.79) | 0.013 * | 3.22 (0.77–13.50) * |
| Minimal Cross-Sectional Area ($mm^2$) | 202.50 (103.03) | 143.75 (103.23) | 0.026 * | 2.96 (0.95–9.21) * |

*3.1. Cephalometric Analysis*

The cephalometric variables were compared between the JIA study group and the control group. Only the variable of the posterior facial height was found to be statistically and significantly different between the two groups, measuring 71.06 mm in the JIA group and 75.39 mm in the control group ($p < 0.05$). The ODDs ratio confirmed the posterior facial height to be significantly smaller in the JIA group.

*3.2. Airway Analysis*

The airway variables were compared between the JIA study group and the control group. The JIA sample was significantly different than the control group for the following airway parameters. The total upper airway volume (nasopharynx and oropharynx combined) was 15,490 $mm^3$ in the JIA group and 19,874 $mm^3$ in the control group ($p < 0.05$). The nasopharynx volume was 4392 $mm^3$ in the JIA group and 5827 $mm^3$ in the control group ($p < 0.01$). The oropharynx volume in the JIA group was 11,098 $mm^3$ compared to 14,047 $mm^3$ in the control group, which was close to but did not quite reach statistical significance ($p = 0.051$). The MCCA was 144 $mm^2$ in the JIA group and 203 $mm^2$ in the control group ($p < 0.05$). The ODDs ratio also indicated a significant association of JIA patients with the variables measured in all the airway analysis.

**4. Discussion**

The JIA and control groups being evaluated were age-matched ($p = 0.825$) to account for normal growth-related changes in cephalometric and airway parameters [52]. The cephalometric values evaluated allowed for the study of the anteroposterior and vertical craniofacial morphologies, which are of interest in this study because there is evidence that anteroposterior jaw positions and mandibular posterior rotation can affect the airway.

*4.1. Cephalometric Parameters*

Many previous studies have documented differences in cephalometric variables between JIA patients and control patients [5,6,13,15,21–27]. However, our data did not demonstrate statistically significant differences in cephalometric values between the two groups in any variable other than posterior facial height ($p = 0.024$). A difference in the posterior facial height has been previously demonstrated in other studies [13,15,23]. One possible explanation for the lack of differences in cephalometric values could be that the control group represented an already retrognathic sample. It does appear that both our JIA and control groups have a tendency toward mild mandibular retrognathia, with the ANB angles of the JIA group and control group measured at 3.21 and 3.69 degrees, respectively. Additionally, the absence of cephalometric differences between the groups may suggest that the treatment regimens employed in the treatment of JIA patients were effective at minimizing craniofacial growth disturbances secondary to JIA.

### 4.2. Airway Parameters

Previous studies have demonstrated that groups with different craniofacial growth patterns have shown differences in CBCT-derived airway parameters [53–57]. Interestingly, despite minimal cephalometric differences between our two groups, we did find statistically significant differences in nasopharyngeal airway volume (NAV), total upper airway volume (TUAV), and MCCA. In the JIA group, 16 of 32 patients (50%) had an MCA of less than 100 mm$^2$, while the control group had only 6 of 32 patients (18.75%) with an MCA less than 100 mm$^2$. The value of 100 mm$^2$ was chosen because it roughly represents the lowest quarter (25%) of the range of MCCA values that were found in both our JIA and control groups, as well as being easy to visually identify using Dolphin Imaging. In addition, the oropharyngeal airway volume (OAV) difference between the groups approached statistical significance ($p = 0.051$). Another possible question that arose from this study was "Could the posterior face height be a significant factor in airway volume and MCCA?". However, this seems unlikely given the difference in posterior face height between the two groups was 4.33 mm, which represents about 5% of the average posterior face height of 71.06 mm in the JIA group. However, the 2D representation of the cephalogram does not take into account the 3D anatomical dentofacial deformity present in some JIA patients (Figure 3). Finally, evidence of sleep-disordered breathing (SDB) has been previously shown in JIA patients that have no evidence of cervical spine or TMJ arthritis [48]. At present, not enough information is available to investigate other reasons as to why airway parameters were different between the two groups. However, based on our data, it does appear that additional factors other than craniofacial morphology may contribute to the anatomic airway reduction in JIA patients. In the general pediatric population, other factors that could contribute to airway compromise include obesity, size of adenoids and tonsils, and presence of allergic rhinitis [42,58–66].

Anatomic and structural features alone are certainly part of the overall pathogenesis of SDB and OSA [60]. However, the fact that patients with OSA do not have airway compromise when awake demonstrates that there is more to the picture than just anatomy, and in fact, upper airway neuromuscular tone plays a large role in normal respiratory patency and function [67]. Additionally, children have been shown to have a reduced tendency toward cortical arousals that would otherwise correct for obstructions that occur during sleep [67]. Nevertheless, correlations between CBCT-derived airway dimensions and AHI have been shown, demonstrating that airway anatomy is a definite factor in airway compromise during sleep and that CBCT should be viewed as a useful tool in evaluating the airway in pediatric patients [34,35]. That being said, CBCT airway analysis certainly has its limitations. A CBCT image provides a representation of the airway at only one static time point of the respiratory cycle with the patient in an upright and awake posture. Additionally, there is evidence that successive CBCT images taken on the same patient only a few months apart are prone to considerable variations in airway volumes [68].

Visual inspection of the 3D airway morphology revealed an interesting finding that requires further investigation from a larger sample. Of the 32 patients with JIA, 21 demonstrated asymmetric deviations of the airway form when evaluated from the 3D frontal view of the airway volume. This represented 67% of the patients reviewed. It is not uncommon for a JIA patient population to also show facial asymmetries when evaluating skeletal and soft tissue forms, but to be able to visualize this in a 3D manner was revolutionary (Figure 4). Further work is required in evaluating for the presence of a correlation between the skeletal/facial and airway asymmetries.

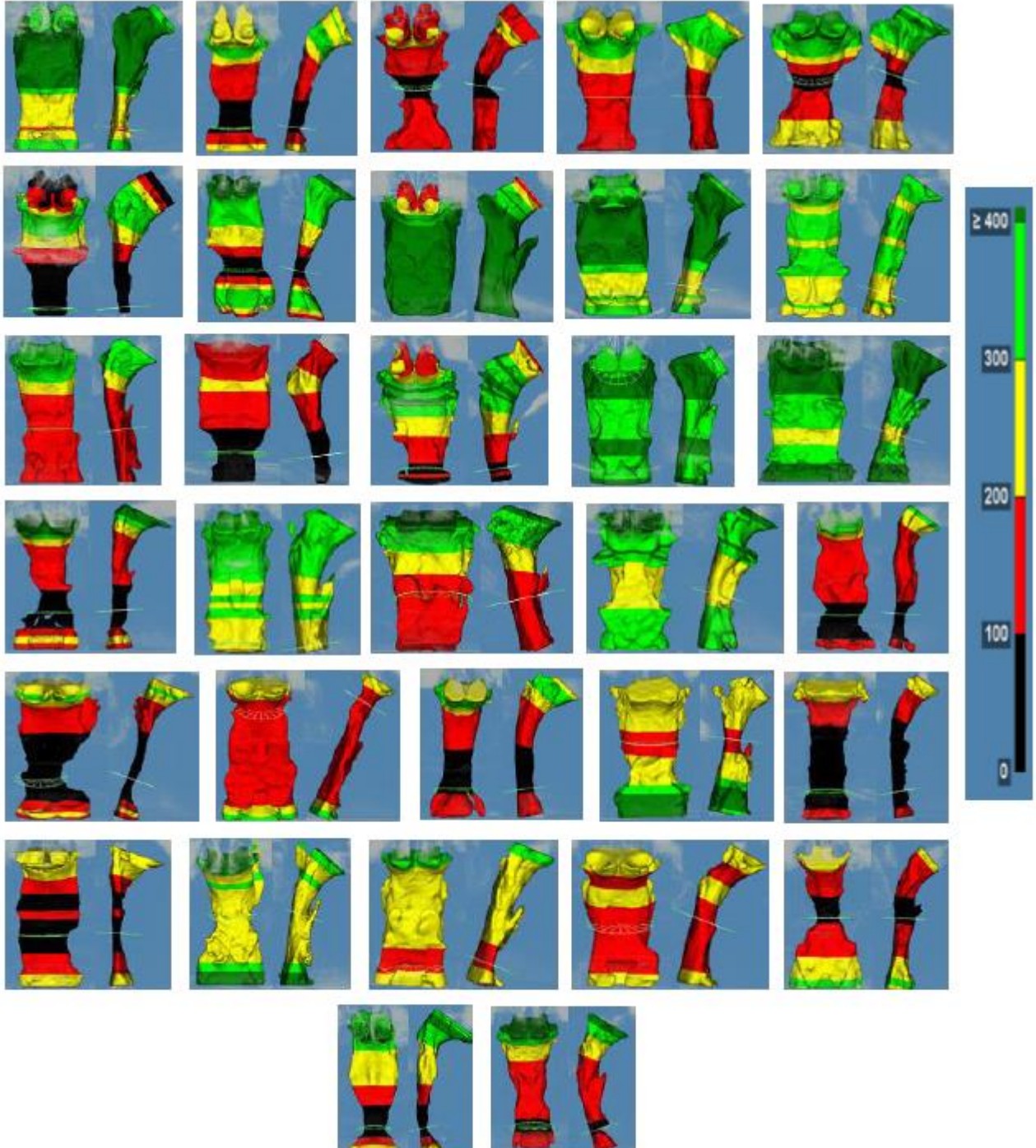

**Figure 3.** Frontal and sagittal views of the airways of all JIA patients with the scale measured in square millimeters of airway cross-section and a marker showing the level of the minimal cross-sectional area of each. Please note that visualizing the color map of the airway utilizes a different feature of Dolphin Imaging, which does not allow for as much precision in setting the upper and lower airway boundaries and is only used here to give a visual representation of the cross-sectional areas at different levels of the airway. All quantitative volumetric and minimal cross-sectional data were gathered with the segmentation boundaries set, as described in Section 2 and as shown in Figure 2.

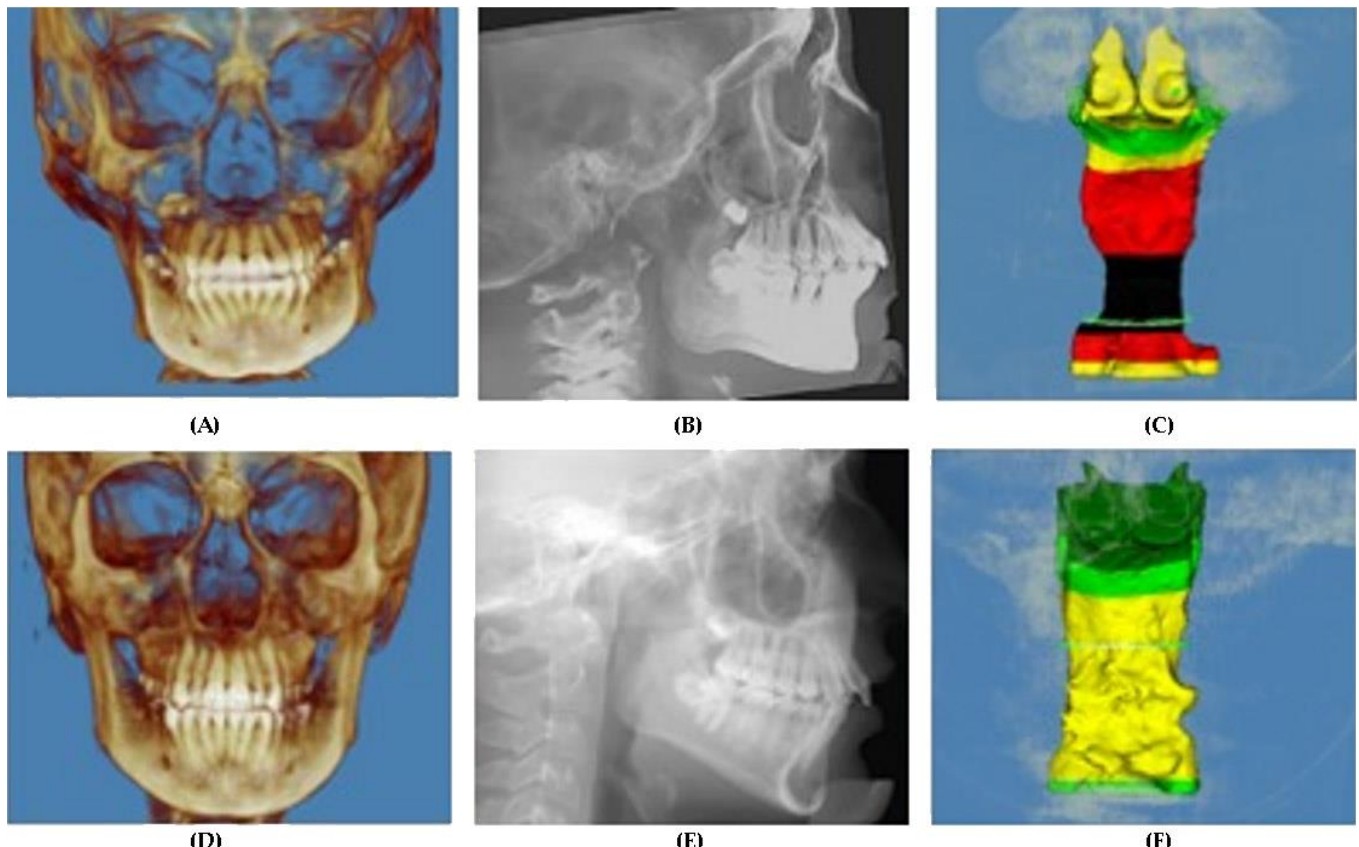

**Figure 4.** (**A–C**) Images from a JIA patient with a chin deviation to the left side and an asymmetric airway with a "twist" effect. (**D–F**) Images of a control patient representative of the control group with no skeletal asymmetry and an airway with no gross asymmetries.

*4.3. Software Analysis and Interpretations*

When evaluating the software programs commercially available for airway analysis of the CBCT DICOM data, some positive and negative qualities have been noted. Dolphin Imaging, which was used in our study, has been validated as having high reliability in measuring the airway consistently between repeated measurements of the same image data and when comparing two airways when both were measured with Dolphin Imaging [69]. For this reason, the data obtained for the JIA and control groups in our study have a high level of validity, given that the airways were all rendered with Dolphin Imaging using the same technique and by the same operator. However, these same software programs, including Dolphin Imaging, have been shown to have poor accuracy when comparing the airway volume values between different software programs [69]. This finding, combined with the fact that there are not currently clear segmentation standards for landmarks used to define the upper and lower boundaries of the different subdivisions of the upper airway, make it difficult to compare airway volumes from one study to another or to use proposed normative values to evaluate study results or individual patients.

*4.4. JIA, Airway and OSA*

Timely diagnosis of TMJ arthritis in JIA patients is critical to allow for early intervention and to reduce the risk of any possible craniofacial growth alterations. Our study demonstrated a reduced airway dimension in JIA patients, and this finding may have an effect on OSA. Due to the significant influences that OSA may have in overall health and disease management, early identification of debilitating craniofacial effects of JIA is critical. Referral for orthodontic evaluation of the craniofacial structures may provide additional information useful in evaluating OSA risk factors. While CBCT images cannot provide a

diagnosis of OSA, they may be a useful screening tool. Additionally, there is evidence that screening JIA patients with the Pediatric Sleep Questionnaire (PSQ) could lead to a timelier diagnosis and treatment of SDB [36].

*4.5. Study Limitations*

Our study is unique in that no one has previously demonstrated 3D airway differences in a JIA patient population. However, our study did have a number of limitations. The retrospective study design does not allow for ideal standardization of the CBCT acquisitions. Not all CBCT images were taken by a single operator, which may have some effect on optimal patient positioning. Additionally, there was no ability to make sure that all patients were given the same instructions for breathing and tongue posture instructions during the image acquisition. Additionally, when considering selection of patients for the control group, it is not standard protocol in our clinic to have a CBCT image taken for all patients undergoing orthodontic evaluation, and CBCT images are often acquired when there is an out-of-the-ordinary finding, dentofacial diagnostic need, or suspicion. For this reason, it is possible that our control group consisted of patients that are not a true representation of the standard norm. During the segmentation of the airway, a common threshold value was used for all patients, while other authors have discussed using an "interactive" method of setting the threshold value in which the threshold is determined for each patient individually at a number that provides a visual best fit of the airway. However, there is no evidence to suspect that our segmentation method created errors on any individual scans.

## 5. Conclusions

The following conclusions were obtained from this study:

(1). There was a difference in the posterior face height between JIA and control patients.
(2). There was a significant difference in total upper airway volume, nasopharynx airway volume, and most constricted cross-sectional area measurements between JIA and control patients.
(3). The ODDs ratio analysis confirmed the association of the statistically significant parameters above.
(4). 50% of JIA patients had an airway with a most constricted cross-sectional area of less than 100 mm$^2$.
(5). 67% of JIA patients had an asymmetric airway form.

**Author Contributions:** Conceptualization, C.H.K.; formal analysis, M.G., M.L.S., T.P. and C.H.K.; investigation, R.Q.C. and B.E.K.; methodology, M.G. and C.H.K.; project administration, T.P. and C.H.K.; writing—original draft, M.G. and C.H.K.; writing—review and editing, R.Q.C. and M.L.S. All authors have read and agreed to the published version of the manuscript.

**Funding:** Biomedical Research Award, American Association of Orthodontics Foundation.

**Institutional Review Board Statement:** University of Alabama at Birmingham Institutional Review Board gave approval for the study. IRB#30000975.

**Informed Consent Statement:** Informed consent was obtained from all patients in the study.

**Data Availability Statement:** Data is available at the UAB Department of Orthodontics repository.

**Conflicts of Interest:** The authors declare no conflict of interest.

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
