# Peer review of "A 3D CBCT Analysis of Airway and Cephalometric Values in Patients Diagnosed with Juvenile Idiopathic Arthritis Compared to a Control Group"

_applsci, doi:10.3390/app12094286_

Round 1

Reviewer 1 Report

As we know that the temporomandibular joint (TMJ) is affected in 30-45%% of Juvenile Idiopathic Arthritis (JIA) patients, with all JIA subtypes at risk for TMJ involvement. JIA patients with TMJ involvement may present with altered craniofacial morphology, including micrognathia, mandibular retrognathia, a hyperdivergent mandibular plane angle, and skeletal anterior open bite. The study comprised of a group of 32 JIA patients and a group of 32 healthy control subjects. CBCT images were taken for all patients and were imported into Dolphin Imaging software. Dolphin Imaging was used to measure the upper airway volumes and most constricted cross-sectional areas of each patient. Cephalometric images were rendered from the CBCT data for each patient and the following cephalometric values were identified: SNA angle, SNB angle, ANB angle, anterior facial height (AFH), posterior facial height (PFH), mandibular plane angle (SN-MP), FMA (FH-MP), overjet (OJ), overbite (OB). Airway volumes, most constricted cross-sectional area values, and cephalometric values were compared between the JIA and control groups. 
At the end of the study, they found that for airway values, statistically significant differences were seen in the nasopharynx airway volume (p=0.004), total upper airway volume (p=0.013), and mostconstricted cross-sectional area (p=0.026). The oropharynx airway volume approached statistical significance (p=0.051). For cephalometric values, only the posterior facial height showed statistically significant difference (p=0.024). 
They concluded that there was a significant difference in the nasopharynx airway volume, total upper airway volumes, and most constricted cross-sectional area of JIA patients compared to healthy controls. In addition to, when comparing cephalometric values, The authors observed that only the posterior facial height was  to be different between the groups.

I think it is a study that offers very interesting and meaningful information. In my opinion, the results obtained are quite interesting for researchers who are interested in this field.

Moreover, the study is well designed and written.

Author Response

Dear Reviewers, thank you for your kind reviewers. We found the findings interesting and hope we can obtain more support in the future to carry out more investigations.

Reviewer 2 Report

Dear Authors, 

I appreciate the efforts put in the current research. However, I would like to suggest modifications/clarifications in the following areas. 

  1. Title of study- Mention "Comparative analysis of ......." or Airway analysis ..... - A case-control study". Also, I feel the title is not descriptive enough, as there is no mention of cephalometric analysis is mentioned. Additionally, the method of intervention, i.e CBCT is not there.... I feel the title needs to be rewritten. 
  2. Abstract section - The term "approaching statistical significance" is not acceptable. The authors should mention the results in an objective way, as "statistically significant" or " non-significant". Authors can mention in a way - that although the results are clinically non-significant, it's crucial or worth discussing. Also, the authors can state the probable reasons for non-significant results in the discussion section. 
  3. Abstract section - the conclusion is more or less the repetition of results. Kindly rephrase and try to highlight the clinical inference. 
  4. Introduction section- line 80, the authors are discussing about the merits and application of CBCT, but cited reference no. 27 which do not justify the text. suggested DOI: 10.21474/IJAR01/2405
  5. Even for retrospective study, ethical approval from the local ethical committee is required. Its not mentioned in methodology 
  6. Add representative Cephalometric image of each study group.
  7. Change the title of Table 1. The present title is a footnote information. 
  8. IN statistical analysis- mention student unpaired t test was used, 
  9. I strongly recommend authors to add stable of ODDS ratio for having OSA for patient with specific cutt off of cephalometeric values. 

Best Regards

Author Response

Dear Reviewer, Thank you for your kind review. We have added the following changes as per your suggestion:

  1. The title has been changed to "A 3D CBCT analysis of skeletal features and airway analysis of patients with juvenile idiopathic arthritis compared to a control group"
  2. We changed the information and removed the term approached to not statistically significant
  3. We changed the conclusion to: There was a significant difference in airway dimensions in the JIA patients as compared to control patients. In addition, the posterior facial dimension seems to be affected in JIA patients.
  4. We removed #27 and added the ref: DOI: 10.21474/IJAR01/2405
  5. We added the ethical approval to the methodology
  6. We added the Cephalometric images to paper
  7. We changed the title of Table 1
  8. We mentioned the unpaired student t-test as suggested
  9. We calculated and added the ODDS ratio

Reviewer 3 Report

Thanks for the opportunity to read your manuscript. I would be grateful for considering the following points:
- On other Carestream models, the exposure time differs from exam to exam. Please verify that the exposure time did not take various amounts and that it is not necessary to add the word "approximately".
- Please add standard information about the parent company in brackets to the name of the Excel program.
- Please verify the wording "Data from the first and second investigators".
I consider the manuscript to be very good and ready for publication after minor corrections.

Author Response

Dear Reviewer,

Thank you for your kind review. Please see our answers to your questions:

  1. We confirm that the CBCT exposure was the same for all subjects. It was 14 seconds.
  2. We added the standard information in the Excel program

Round 2

Reviewer 2 Report

Dear Authors, 

I can see the improvement in the manuscript, but a few areas are still unattended. I would suggest authors understand the comments and incorporate them. 

  1. The odds ratio is represented in specific numbers such as 1.13 or 0.96, as the number will reflect the likelihood of having risk in case the patient has an underlying risk factor. Also, the ODD ratio is mentioned along with the confidence interval (CI). Kindly modify the table accordingly. The authors have not discussed the findings of the ODD ratio in the discussion, conclusion and abstract section. 
  2. The authors have not phrased the statement of title 1. It can be written as "Comparative analysis of airway and cephalometric parameters among the study groups". In the footnotes, mention the abbreviation used in the tables. The mention of statistical tools will be in the material and methods section - "Statistical analysis". 

Best Wishes

Author Response

Dear Reviewer, Thank you so much for the kind reviewer. We calculated the ODDs ratio and indeed we found the association to be similar to our earlier. This gives us good confidence in our findings and I want to thank you for teaching us an alternative way of expressing the data. Your time is much appreciated.